# Influence of Nitrogen-Modified Atmosphere Storage on Lipid Oxidation of Peanuts: From a Lipidomic Perspective

**DOI:** 10.3390/foods13020277

**Published:** 2024-01-16

**Authors:** Xia Ma, Wenhao Li, Huayang Zhang, Peng Lu, Pengxiao Chen, Liang Chen, Chenling Qu

**Affiliations:** 1School of Food and Strategic Reserves, Henan University of Technology, Zhengzhou 450001, China; maxia2022@163.com (X.M.); liwenhao0730@163.com (W.L.); zhy121100@163.com (H.Z.); 18737737792@163.com (P.L.); cpx2020@haut.edu.cn (P.C.); 2School of Biological Engineering, Henan University of Technology, Zhengzhou 450001, China

**Keywords:** peanut, lipid oxidation, nitrogen-modified atmosphere storage, non-targeted lipidomics

## Abstract

The effect of nitrogen-modified atmosphere storage (NS) on peanut lipid oxidation was investigated in this paper. Non-targeted lipidomics was employed to detect the lipid metabolites in peanuts with the aim of exploring the mechanism of lipid oxidation in peanuts under different storage conditions. The results showed that compared with conventional storage (CS), NS significantly (*p* < 0.05) delayed the increase in acid value, carbonyl value, and 2-thiobarbituric acid value and the decrease in vitamin E content. However, the storage time has a much greater effect on lipid oxidation than the oxygen level in the storage environment. Lipidomics analysis revealed that there were significant differences in metabolite changes between CS and NS. NS reduced the decline of most glycerophospholipids by regulating lipid metabolism in peanuts. NS maintained higher levels of Diacylglycerol (DAG), sulfoquinovosyl diacylglycerol (SQDG), lysophophatidylcholine (LPC), lysophosphatidylethanolamine (LPE) and phosphatidylinositol (PI) compared to CS. This work provided a basis for the application of NS technology to peanut storage.

## 1. Introduction

The peanut is a widely grown oilseed crop in the world [1]. It is a good source of protein and lipids, which are beneficial to human health, e.g., reducing blood pressure and cholesterol to prevent cardiovascular diseases [2,3]. Peanuts are used for producing peanut oil and peanut butter in the food industry [2]. Meanwhile, due to their high lipid content, peanuts are prone to oxidative rancidity during storage [4]. Lipid oxidative rancidity could produce aldehydes and ketones, which generate undesirable flavor and quality deterioration of the peanuts [5,6]. It has been reported that advanced lipid oxidation end products are harmful to humans, which are being implicated in diseases such as atherosclerosis, cancer, inflammation, and aging processes [7].

The storage temperature and humidity play important roles in affecting the quality of the peanuts during storage. The peroxide value, carbonyl value, and malondialdehyde content of the peanuts (stored at 15 °C, 25 °C, and 35 °C for 320 days) were significantly increased during storage, and the higher the storage temperature, the higher the degree of lipid oxidation and nutrient loss [8]. High humidity (above 70% relative humidity) can lead to an increase in the moisture content of the peanuts, which results in increased metabolism and the respiratory activity of the peanut seeds and leads to accelerated deterioration of the peanuts [9].

In addition, the gas composition in the storage environment is also an important influencing factor. Modified atmosphere storage (MAS) was considered an efficient approach to maintaining the favorable quality of various products by reducing the concentration of oxygen [3]. It was reported that zero-oxygen hermetic packaging could help suppress aflatoxin production and quality deterioration of the peanuts [10]. It was observed that the nitrogen-modified packaging of fresh edible peanuts delayed the browning of the kernel, protected cell membrane integrity, and inhibited the loss of antioxidant substances [4]. The levels of both mycotoxigenic fungi and wheat pests were significantly reduced under the nitrogen or carbon dioxide-modified atmosphere storage [11,12]. The carbon dioxide-modified atmosphere storage delaying the deterioration of paddy rice was also reported [13].

Lipidomics was a branch of metabolomics that allowed the characterizing and quantifying of lipids in biological samples [14]. In recent years, lipidomics based on LC-MS technology have been widely used in the analysis of various oil-rich products owing to their high separation capacity, high throughput, high sensitivity, and high resolution [15]. A previous study analyzed lipids in different varieties of rice to explore further the contribution of lipids to different rice traits with non-targeted lipidomics. Significant lipid differences were observed among different rice samples, and a strong correlation was also observed between lipids and rice textural properties [16].

The safe storage of peanuts was important for the stable supply of peanut products in the market and their edible safety. The aim of this study was to investigate the effects of conventional storage (CS) and nitrogen-modified atmosphere storage (NS) on peanut-lipid oxidation and peanut-lipid metabolism.

## 2. Materials and Methods

### 2.1. Materials and Storage Condition

Peanuts (Wanhua 2) were harvested in Fangcheng City, Henan Province, China. The peanuts were manually shelled, and the obtained peanut kernels were used as raw material for the storage experiments.

All reagents were commercially available and were used as purchased. Petroleum ether, benzene, and ethanol were purchased from Tianjin Tianli Chemical Reagent Co., Ltd., Tianjin, China. Moreover, 2,4-dinitrophenylhydrazine and potassium hydroxide were purchased from Tianjin Komeo Chemical Reagent Co., Ltd., Tianjin, China. Furthermore, 2-thiobarbituric acid was purchased from Shanghai Sinopharm Group Chemical Reagent Co., Ltd., Shanghai, China. All reagents were of analytical grade and used without any further purification. 

Two storage conditions, conventional storage (CS) and nitrogen-modified atmosphere storage (NS), were used in our experiment. Conventional storage (CS): the peanut kernels were packed in a transparent bag made of high-polymer nylon fabric and composite polyethylene (25 cm × 36 cm in length and width, and 0.16 mm in thickness) and stored in an incubator (Ningbo Southeast Instrument Co., Ltd., Ningbo, China) for 300 days at 35 °C. Nitrogen-modified atmosphere storage (NS): peanut kernels were packed in the same kind of bag as CS, and the bag was filled with high nitrogen to the final concentration of 98 ± 0.5%. Then, it was stored in the incubator for 300 days at 35 °C.

### 2.2. Determination of Peanut-Lipid Oxidation Properties

#### 2.2.1. Extraction of Peanut Lipid

The lipid extraction was performed according to the methods mentioned in the literature [17]. Peanut kernels were first sliced into 1 mm slices. Then petroleum ether was added for lipid extraction for 8 h. After extraction, the mixture was filtered, and the filtration was evaporated using RE2000A rotary evaporator (Shanghai Yarong Biochemical Instrument Co., Ltd., Shanghai, China) at a negative pressure and a water bath temperature of less than 40 °C. The distillation residue was then used for analysis.

#### 2.2.2. Measurement of Acid Value

The acid value (AV) of oil was determined according to the literature [18]. Moreover, 0.5 g oil sample was dissolved in 50 mL ethanol, and the solution was shaken well. The solution was then titrated with standard potassium hydroxide solution using phenolphthalein as the indicator. The AV was expressed in mg KOH/g of sample. AV was calculated according to the following equation:(1)$$AV=\frac{56.1\cdot C\cdot \left(V-V_{0}\right)}{m}$$
where C was the normality of the potassium hydroxide solution, V was the volume of the potassium hydroxide solution consumed in the sample test, V_0_ was the volume of the potassium hydroxide solution consumed in the blank test, and m was the weight of the sample.

#### 2.2.3. Measurement of Carbonyl Value

The carbonyl value (CV) was determined following the method mentioned in the literature [8]. The oil was first dissolved in benzene in a tube, and then 2,4-dinitrophenylhydrazine was added to it. The tube was heated in the water bath at 60 °C for 30 min and cooled to room temperature. Ethanol solution of potassium hydroxide was added to the tube. The absorbance was measured at 440 nm. CV was calculated according to the following equation:(2)$$CV=\frac{A}{854\cdot m}\cdot 1000$$
where A was the absorbance of the sample solution and m was the weight of the sample.

#### 2.2.4. Measurement of 2-Thiobarbituric Acid Value

The 2-thiobarbituric acid value (2-TV) was determined based on the literature [19]. The oil sample was mixed with 2-thiobarbituric acid solution. The absorbance of the mixture was measured at 530 nm. Moreover, 2-TV was calculated according to the equation:(3)$$2-TV=\frac{50\cdot \left(A-A_{0}\right)}{m}$$
where A was the absorbance of the sample solution, A_0_ was the absorbance of blank, and m was the weight of the sample.

#### 2.2.5. Measurement of the Content of Vitamin E

The content of vitamin E (VE) for lipids was determined using a colorimetric kit (Kit No. A008-1-1, Nanjing Jiancheng Institute of Biological Engineering, Nanjing, China). VE had the ability to reduce Fe^3+^ to Fe^2+^ in the presence of phenanthroline. Fe^2+^ could form pink complex with phenanthroline; the pink complexes were strongly absorbed at 533 nanometers. The content of VE was determined according to the calibration curve y = 1.9246x − 6.2195, R^2^ = 0.9837.

### 2.3. Detection and Analysis of Metabolites

#### 2.3.1. Extraction of Metabolites

The extraction of metabolites was based on the methods in the literature [20]. Moreover, 25 mg of peanut sample was extracted with 480 μL extract solution (methyl tert-butyl ether: methanol = 5:1). The mixture was homogenized at 35 Hz for 4 min and sonicated in an ice-water bath for 5 min, and the homogenization and sonication processes were repeated 3 times. Then, the sample was incubated at −40 °C for 1 h and centrifuged at 3000 rpm for 15 min at 4 °C. After centrifugation, 300 μL of the supernatant was dried under vacuum. After drying, 200 μL of the solution (dichloromethane:methanol = 1:1) was added to re-dissolute the dried sample, which was vortexed for 30 s and sonicated in an ice water bath for 10 min. The sample was then centrifuged at 4 °C for 15 min at 13,000 rpm, and 120 μL of the supernatant was used for test. The quality control (QC) sample was prepared by mixing an equal aliquot of 10 μL of the supernatants from all the samples. The obtained supernatant and QC samples were used for subsequent LC-MS analysis.

#### 2.3.2. LC-MS Analysis

LC-MS analysis was performed using Agilent 1290 (Agilent Technologies, Beijing, China) equipped with phenomen kinetex C18 (2.1 × 100 mm, 1.6 μm) liquid chromatographic column, which was employed for chromatographic separation of target compounds. The mobile phase A consisted of 40% water and 60% acetonitrile solution with 10 mmol/L ammonium formate, the mobile phase B was composed of 10% acetonitrile and 90% isopropanol solution (50 mL of 10 mmol/L aqueous ammonium formate solution was added per 1000 mL). The analysis was carried out with elution gradient as follows: 0~1.0 min, 40% B; 1.0~12.0 min, 40%~100% B; 12.0~13.5 min, 100% B; 13.5~13.7 min, 100%~40% B; 13.7~18.0 min, 40% B. The analysis was performed under the conditions of the mobile phase flow rate of 0.3 mL/min, column temperature of 55 °C, and sample tray temperature of 4 °C. Furthermore, both negative and positive ion injection volumes were 2 μL.

The primary and secondary mass spectrometry data were acquired using the Thermo Q Exactive Orbitrap mass spectrometer (Thermo Fisher Scientific, Waltham, MA, USA) under the control of acquisition software (Xcalibur, version 4.0.27, Thermo). The electrospray ionization (ESI) source conditions were set as follows: sheath gas flow rate as 30 Arb, aux gas flow rate as 10 Arb, capillary temperature as 320 °C (positive) or 300 °C (negative), full ms resolution with 70,000, MS/MS resolution with 17,500, collision energy as 15/30/45 in secondary fragmentation energy mode, spray voltage as 5 kV (positive) or −4.5 kV (negative).

#### 2.3.3. Identification of Metabolites

The raw data files were converted to files in mzXML format using ‘msconvert’ program of ProteoWizard software 3.0. Retention time correction, peak identification, peak extraction, peak integration, and peak alignment were then implemented using the XCMS algorithm. The minfrac for annotation was set at 0.5, and the cutoff for annotation was set at 0.3. Lipid identification was achieved with LipidBlast library, which was developed using R package and based on XCMS. The identification of metabolites was based on MS/MS secondary mass spectrometry data (Appendix A). The secondary mass spectra matching qualitative scoring values, retention time, and mass-to-charge ratio of the metabolites are shown in Appendix A. The peak areas of the metabolites were used for the quantification of the metabolites.

### 2.4. Statistical Analysis

One-way ANOVA was performed using SPSS 24 for peanut-lipid oxidation characteristics, and Duncan multiple test model was used for significance comparison, with *p* < 0.05 to meet the significance requirement. The raw lipidomics data were first preprocessed, including deviation filtering, missing value filling, and normalization. Then, comparative analysis of lipid metabolites between peanut samples was performed using principal component analysis (PCA) and orthogonal partial least squares discriminant analysis (OPLS-DA) with Metaboanalyst 5.0. Data from different experimental groups were screened for differential metabolites according to specific criteria (*p* < 0.05, VIP > 1). The lipid metabolic pathway analysis was performed according to the Kyoto Encyclopedia of Genes and Genomes (KEGG).

## 3. Results and Discussion

### 3.1. Quality Changes of Lipid Extracted from Peanuts during Storage

The AV represented the degree of lipid rancidity. It can be seen from Figure 1A that the AVs of lipids extracted from peanuts stored under different conditions gradually increased along with the storage time. The AVs for those under NS were significantly lower (*p* < 0.05) than those under CS at the same storage time. 

The CV and 2-TA indicated the level of secondary oxidation products. During the storage of peanuts, primary lipid oxidation products (hydroperoxides) can convert into secondary products, leading to an increase and then a decrease in the peroxide value (reflecting the amount of the primary oxidation products) [8]. Therefore, monitoring the amount of secondary oxidation products could better reflect the oxidation of peanut oil under different storage conditions. The secondary oxidation products that contain carbonyl groups were more stable than hydroperoxides, so the CV was a good indicator of lipid oxidation [21]. Malondialdehyde (MDA) concentration indicates lipid peroxidation of membranes, which can be examined via 2-TA assay [22,23]. It can be seen from Figure 1B,C that the CVs and 2-TAs all displayed a rising tendency along with the duration of storage. The CVs and 2-TAs for those under NS remained at a lower level compared with CS. In addition, there were significant differences (*p* < 0.05) in 2-TAs of peanuts under different storage conditions at the end of the storage. Therefore, from the results shown in Figure 1B,C, the content of the secondary oxidation products increased along with storage, and NS could slow down the secondary oxidation of peanuts.

VE was an antioxidant present in plant seeds. It could inhibit the propagation of the lipid oxidation chain reaction [24]. As shown in Figure 1D, the VE contents of lipids from peanuts stored in various conditions showed overall decreasing trends during storage. The VE contents for those under NS decreased more slowly than those under CS. It suggested that NS can reduce VE loss in peanuts.

In summary, the peanut-lipid oxidation during storage was investigated mainly from the primary oxidation, secondary oxidation, and antioxidant perspective in this part. The results showed that the degree of lipid oxidation intensified with the prolongation of storage time and that NS could delay lipid oxidation in peanuts to a certain extent. However, the storage time has a much greater effect on lipid oxidation than the oxygen level in the storage environment.

### 3.2. Overview of Lipidomic Profiles for Lipid Extracted from Peanuts

To investigate the changes in peanut lipids after storage under different storage conditions, the lipid metabolites were determined using non-targeted lipidomics. Principal component analysis (PCA) was first conducted to reveal the overall differences in lipid metabolites [25]. It was shown in Figure 2A that the first and second principal components explained 27.5% and 20.4% of the total variation of lipids in peanut samples at the initial storage (0 d), under CS for 300 d (CS 300 d), and under NS for 300 d (NS 300 d), with all samples falling within the 95% confidence interval. However, PCA did not reveal differentiation among these three samples. Therefore, orthogonal projections to latent structures-discriminant analysis (OPLS-DA) were employed to analyze the differences between samples of diverse groups. The score scatter plots of OPLS-DA (Figure 2B,C) showed a clear separation between CS 300 d and 0 d, and between NS 300 d and CS 300 d.

To maximize the differentiation of significantly different metabolites, all lipid metabolites were screened with VIP > 1.0 and *p* < 0.05. The screening results were rendered in the form of volcano plots with the log2(FC) of various substances and −log10p (Figure 2D,E), which can visually demonstrate the overall distribution of metabolite differences between different samples. In the volcano plots, each point represented one metabolite; red points denoted the significantly up-regulated metabolites, blue points represented the significantly down-regulated metabolites, and gray points referred to the non-significant regulated metabolites.

The results showed that there were 186 significant lipid metabolites in the CS 300 d vs. 0 d group, where 95 metabolites were down-regulated and 91 metabolites were up-regulated (Figure 2D and Appendix A). In the NS 300 d vs. CS 300 d group, 28 significant lipid metabolites were found, with 18 metabolites down-regulated and 10 metabolites up-regulated (Figure 2E and Appendix A). And hierarchical clustering heat maps (Figure 2F,G) were used to visualize the lipid content changes between different groups based on the 25 top significant ion features marked with red (up-regulated) and blue (down-regulated).

The above lipidomic results indicated the significant lipid metabolite changes for peanuts between the CS 300 d vs. 0 d group and the NS 300 d vs. CS 300 d group. And it re-confirmed that the effect of storage time on lipid oxidation was greater than the oxygen level in the storage environment. To further explore the effect of storage time and storage conditions on the peanut-lipid metabolism, the lipid metabolic pathways were analyzed.

### 3.3. Metabolic Pathway Analysis

The metabolites with significant differences (VIP > 1 and *p* < 0.05) were retrieved with MetaboAnalyst 5.0 for lipid metabolic pathway analysis using the KEGG database (Arabidopsis thaliana). The lipid metabolic pathway analysis between different samples was displayed in bubble plots (Figure 3). Four main metabolic pathways were affected by the storage time of peanuts, including glycerophospholipid metabolism, glycerolipid metabolism, linoleic acid metabolism, and arachidonic acid metabolism. And glycerophospholipid metabolism and glycerolipid metabolism were influenced by the oxygen level in the peanut storage environment. Furthermore, the conversion of lipid metabolites was mapped to show their interrelationship to reveal the mechanism of metabolic differences (Figure 4).

### 3.4. Glycerolipid Metabolism

Glycerolipid is a class of lipids that exists in peanuts, which mainly includes triacylglycerol (TAG) and Diacylglycerol (DAG). TAGs were major storage lipids in seeds, which comprised glycerol and three fatty acid moieties [26]. TAGs served as important fatty acid reserves for carbohydrate and energy production during oilseed development [27]. Glycerolipid metabolism primarily involved the hydrolysis of TAGs by lipases [28]. It was observed that TAGs had no significant differences in the two comparison groups. Due to that, the amounts of TAGs hydrolyzed during storage were much smaller than the total amounts.

DAG was one of the major lipid subclasses in peanut seeds, which was a precursor in the biosynthesis of TAGs and glycerophospholipids [29,30]. It can be seen from Figure 4 that the content of DAGs was down-regulated in the CS 300 d vs. 0 d group, indicating that DAGs had been depleted, decomposed, or converted to other substances. However, there was no significant change in the DAG content of peanuts under NS conditions compared to 0 d. It suggested that NS inhibited further DAG reactions in peanuts, which was consistent with the quality results in Section 3.1.

Diacylglyceryl trimethylhomoserine (DGTS) and sulfoquinovosyl diacylglycerol (SQDG) are two important membrane lipids. DGTS possessed a positively charged trimethylammonium group and a negatively charged carboxyl group (a zwitterionic lipid) [31]. It was found that plants showed a sharp increase in DGTS levels under phosphorus deficiency stress, which was due to the fact that DGTS was analogous to PC and PE, making it ideal for replacing PC and PE under phosphorus-limited conditions [32]. It was reported that SQDG modulates physiological regulatory systems and improves plant tolerance to various abiotic stresses [33]. In our study, DGTS and SQDG all increased after 300 days of storage regardless of storage conditions. It was hypothesized that peanuts triggered protective mechanisms due to environmental stresses during storage, leading to changes in membrane composition.

### 3.5. Glycerophospholipid Metabolism

Although phospholipids accounted for only 1% of total lipids, they had critical biological functions in cell signaling and maintenance of cell membrane integration compared with glycerolipids [34]. Glycerophospholipids mainly included lysophophatidylcholine (LPC), lysophosphatidylethanolamine (LPE), phosphatidylcholine (PC), phosphatidic acid (PA), phosphatidylglycerol (PG), phosphatidylethanolamine (PE), phosphatidylserine (PS), and phosphatidylinositol (PI), etc. [35].

PA was a minor class of phospholipids accounting for about 1% of the total glycerophospholipids. It acts as a precursor of all glycerolipids in the endoplasmic reticulum and can be synthesized either with the hydrolysis of phospholipids by phospholipase D or with the DAG phosphorylation by the DAG kinase [36]. PA was an important lipid second messenger in response to a variety of (a)biotic stresses, where it is linked to various cellular processes, like vesicular trafficking, membrane fission and -fusion, and transport [37]. As shown in Figure 4, the PA of peanuts under NS presented a lower level than 0 d and CS. It indicated that blocking membrane lipid metabolism under NS by limiting the formation of PA may retard peanut aging [38].

PI was another important phospholipid, which could be phosphorylated to seven specific derivatives. These derivatives are involved in a vast array of cellular functions, including signaling, ion channel modulation, and membrane traffic [39]. It was reported that PI decreased by 8.81% in soybean seeds aged 18 d [40], which indicated that the decrease in PI was a sign of seed aging. It was observed that PI was down-regulated in CS 300 d compared to 0 d and NS 300 d, where there was no significant change between 0 d and NS 300 d. This suggested that CS led to the aging of peanuts, whereas NS significantly retarded the aging.

PC was the most abundant phospholipid, which served as a cell membrane structural component to maintain the homeostasis of intracellular phospholipid content [41,42]. It was noted that the excessive accumulation of ROS led to lipid peroxidation, decreased phospholipid content, and exacerbated cell membrane damage [40]. As shown in Figure 4, PC was down-regulated in CS 300 d compared to 0 d and NS 300 d but did not change significantly in NS 300 d vs. CS 300 d. Therefore, we speculated that peanut cell membrane damage occurred due to lipid peroxidation after storage, and NS delayed the lipid oxidation, which was in accordance with the results in Section 3.1.

LPC was one of the most characterized phospholipids, which played a key role in signaling and in the stress response of plants. Stresses such as low temperature and pathogens were reported to cause elevated levels of LPCs [43]. The contents of LPCs increased in both CS and NS conditions, which suggested the stress of long-term storage on LPCs.

LPE was an intermediate in phospholipid metabolism and turnover. It was produced by phospholipase A (PLA) mediated hydrolysis of acyl groups of a phospholipid to lysophospholipids and fatty acids [44]. It was reported that LPE can inhibit the activity of phospholipase D (PLD), which was one of the main enzymes responsible for membrane phospholipid degradation, leading to membrane breakdown and seed senescence [45]. Thus, LPE may retard peanut senescence by inhibiting PLD activity. Our results showed that the content of LPE decreased significantly after CS for 300 days. However, compared to CS, LPE was up-regulated under NS. These results suggested that the decrease in LPE led to peanut aging, which was alleviated under NS.

In summary, most of the phospholipids, including PA, PI, PC, and LPE, decreased in CS after 300 days when compared with 0 d. It was reported that the decrease in phospholipids might lead to the production of oxidative products [28]. These changes in PA, PI, PC, and LPE in peanuts after storage were consistent with the results of Section 3.1. And the changes of PI, PC, and LPE in peanuts under NS demonstrated that NS is beneficial in retarding lipid oxidation.

## 4. Conclusions

In this paper, the effect of nitrogen-modified atmosphere storage (NS) on lipid oxidation in peanuts was investigated. And the mechanism of the NS retarding lipid oxidation was revealed by analyzing the changes in lipid metabolites. The results showed that NS retarded the oxidation of peanut lipids, but the effect of NS on lipid oxidation in peanuts was much less than that of storage time. From lipidomics analysis, it was found that lipid oxidation during storage and the NS retardation of lipid oxidation are mainly regulated by phospholipid metabolism and glycerophospholipid metabolism. In phospholipid metabolism, NS inhibited the subsequent reactions of DAG in peanuts, and the up-regulation of DGTS and SQDG after 300 days of storage indicated the cell membrane composition changed due to the environmental stresses during storage. In glycerophospholipid metabolism, the decrease in PA, PI, PC, and LPE in conventional storage (CS) may lead to the production of oxidative products, and the up-regulation of PI, PC, and LPE in the NS vs. CS group demonstrated that NS was beneficial in retarding lipid oxidation. Therefore, applying nitrogen-modified atmosphere technology to peanut storage can effectively delay lipid oxidation in peanuts and maintain the quality of the peanuts. The present study elucidated the linkage between lipid metabolism and lipid oxidation at the molecular level, which provided a reference for peanut storage.

## Figures and Tables

**Figure 1 foods-13-00277-f001:**
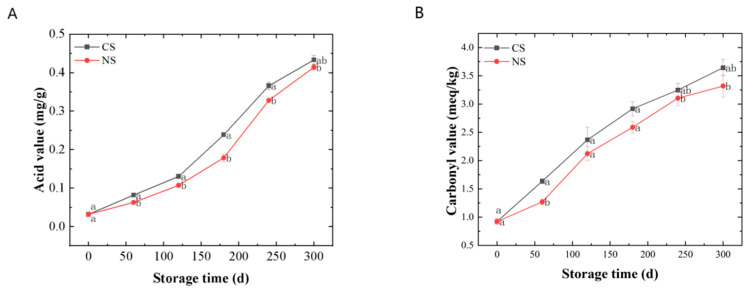
Quality changes of lipids extracted from peanuts during storage. (**A**) acid value, (**B**) carbonyl value, (**C**) 2-thiobarbituric acid value, (**D**) vitamin E. Different letters (a, b) indicated significant differences at the *p* < 0.05 level.

**Figure 2 foods-13-00277-f002:**
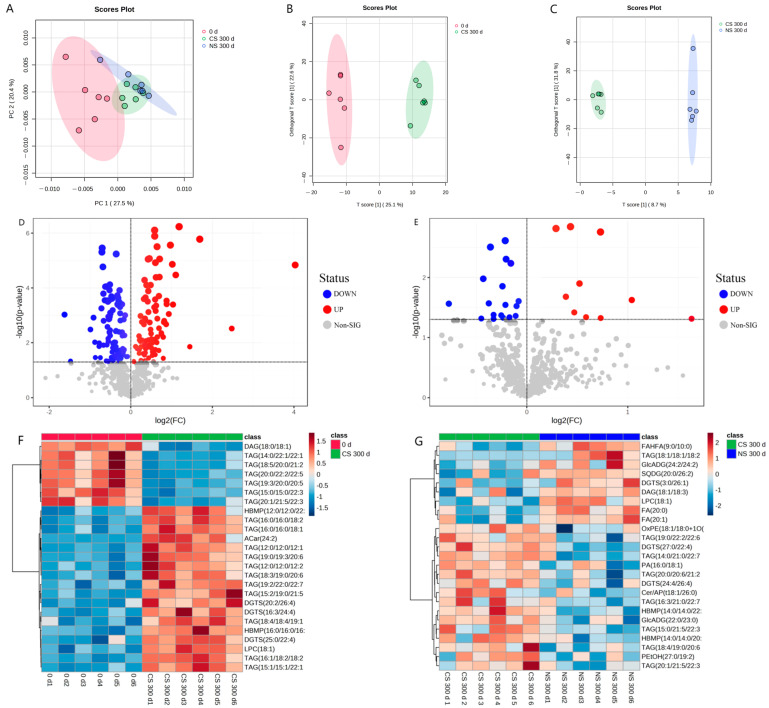
Statistical analysis of lipids extracted from peanuts under different storage conditions and periods using Metaboanalyst 5.0 software: (**A**) PCA score plots of initial point of storage (0 d), CS 300 d and NS 300 d, (**B**) OPLS-DA score plots of CS 300 d vs. 0 d, (**C**) OPLS-DA score plots of NS 300 d vs. CS 300 d, (**D**) Volcano plots of CS 300 d vs. 0 d, (**E**) Volcano plots of NS 300 d vs. CS 300 d, (**F**) Hierarchical clustering heat map of CS 300 d vs. 0 d, (**G**) Hierarchical clustering heat map of NS 300 d vs. CS 300 d.

**Figure 3 foods-13-00277-f003:**
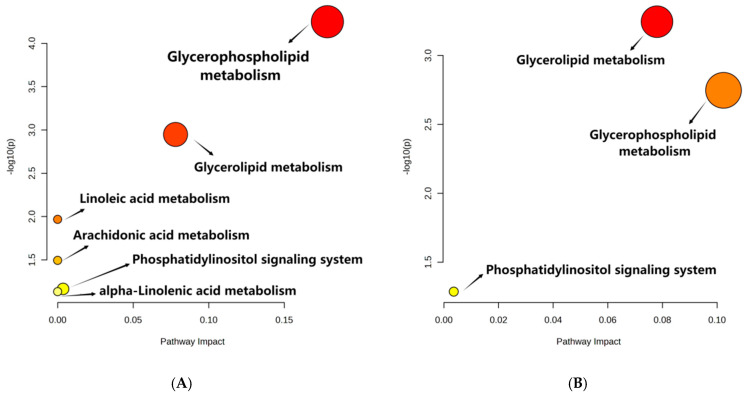
Metabolic Pathways of (**A**) CS 300 d vs. 0 d and (**B**) NS 300 d vs. CS 300 d. Note: CS 300 d vs. 0 d meant metabolites of peanuts under CS after 300 days compared with that at the initial storage. NS 300 d vs. CS 300 d meant metabolites of peanuts under NS after 300 days compared with those of CS after 300 days. The *X*-axis represents pathway impact, and the *Y*-axis represents the pathway enrichment. The redder the color of the bubble, the smaller the *p* value of the metabolic pathway, and the yellower the color of the bubble, the larger the *p* value of the metabolic pathway.

**Figure 4 foods-13-00277-f004:**
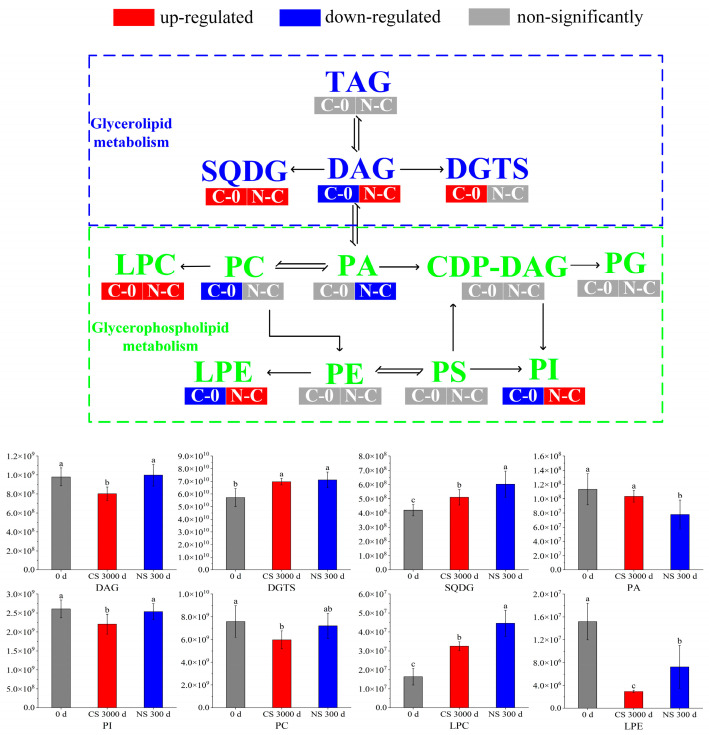
Transformation and metabolic pathways of important metabolites under different storage conditions. Different letters (a, b, c) indicated significant differences at the *p* < 0.05 level.

## Data Availability

Data is contained within the article and Appendix A.

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
