# Peer review of "Influence of Nitrogen-Modified Atmosphere Storage on Lipid Oxidation of Peanuts: From a Lipidomic Perspective"

_foods, 2024, doi:10.3390/foods13020277_

Round 1
Reviewer 1 Report
Comments and Suggestions for Authors
The manuscript is interesting and novel. The title of the manuscript is consistent with the objectives, the work is correctly justified, and the materials and methods used are sufficient to meet the objective. Appropriate statistical analysis was used, and the results obtained were presented and discussed correctly. The conclusion is consistent with the results.
Dear authors, it is necessary to address the following recommendations:
Line 38: use the non-abbreviated form of RH, this is the first time it appears in text
Line 52: According to the author guide (Microsoft word template) it is necessary to avoid using names or surnames of authors in the text, it is only necessary to use the reference number in square brackets. Make the required changes through the document where required.
Line 57-61: It is recommended in this section to only mention the objective of the work, how the experiment will be carried out should be indicated in the materials and methods section.
Line 59: indicate in the text the meaning of CS and NC, although they are already in the summary it is important to indicate them from the beginning of the body of the document (introduction).
Line 63: include a section on reagents and materials where the origin of each one used is included. Also indicate the purity of the materials
Line 67: rewrite… For CS, the peanut…
Line 69: rewrite… For NS, peanut…
Line 74: rewrite… Extraction of peanut lipid
Line 76: rewrite… pieces and petroleum
Line 77: note… include information on the equipment used (model, brand, country) throughout the document
Line 81: rewrite… Measurement of acid value
Line 82-87: the reference numbering sequence was lost
Line 89: rewrite… Measurement of peroxide value
Line 90-97: the reference numbering sequence was lost
Line 99: rewrite… Measurement of carbonyl value
Line 100-105: rewrite… Measurement of peroxide value
Line 102: rewrite… 30 min
Line 107: rewrite… Measurement of p-anisidine value
Line 108-107; 116-119; 122-125; 122-125: The reference number was not included.
Line 115: rewrite… Measurement of 2-thiobarbituric acid value
Note: in the techniques where calibration curves were carried out to quantify compounds, it is necessary to include the equation of the curve obtained, as well as the r2 obtained, and indicate the standard used
Line 121: rewrite… Measurement of vitamin E
Line 128: rewrite… Extraction of metabolites
Line 136: 4°C
Line 141: rewrite… LC-MS analysis
Line 159: rewrite… Identification of metabolites
Line 179,183,191,224, review other cases in the document: abbreviations should be used from the first time they appear in the text, in this case in the materials and methods section, and subsequently, only use the abbreviation throughout the text.
Note: could indicate what the limit values are for each of the metabolites analyzed, to indicate that the product is no longer suitable for consumption.
Line 366: use a dot at the end of the figure title
References section
Note: use lowercase text format for titles of references, except for the first letter of the first word of the title
Note: the name of the reference journal must use italic text formatting
Note: scientific names should be written in italic text format
Reviewer 2 Report
Comments and Suggestions for Authors
the analysis of food and its impact on health is a fast-developing field. the authors choose very interesting and highly consumable food such as peanuts. they responded to the research framework and presented obtained results very well but there is a few points for improvement for example:
in line 41 after the name of the author the reference number should be given not at the end of sentences. the lipidomics if briefly described I suggest the author to improve it for example following papers could help https://doi.org/10.3390/ijms22189698
The introduction tinkles the reader about the research framework but it should give more information on why the authors chose peanuts and analyzed the lipid profile of it. the methodology summarizes all techniques but doesn't give a description of it such as LOD, LOQ repeatability, and linearity ..... the results are presented well, and easy to understand the number of figures and tables is adequate for the presented research. the conclusions are supported by the results. the references are nicely selected but should be improved by a better explanation of lipidomcs and important properties of peanuts.
Comments on the Quality of English Languagethe paper is easy to read and understand the style and grammar are satisfactory.
Reviewer 3 Report
Comments and Suggestions for Authors
The revision is attached in a separate document.

Round 2
Reviewer 2 Report
Comments and Suggestions for Authors
the authors made great effort to improve the paper according to the suggestions and the implemented changes improved the quality of the article and pointed out all the benefits of the peanuts and their lipid profile.
Reviewer 3 Report
Comments and Suggestions for Authors
The manuscript was sufficiently revised by the Authors. I also appreciated resonses on all comments.
